# Soil Health and Arthropods: From Complex System to Worthwhile Investigation

**DOI:** 10.3390/insects11010054

**Published:** 2020-01-16

**Authors:** Cristina Menta, Sara Remelli

**Affiliations:** Department of Chemistry, Life Sciences and Environmental Sustainability, University of Parma, Viale delle Scienze 11/A, 43124 Parma, Italy; sara.remelli@unipr.it

**Keywords:** soil biodiversity, microarthropods, macroarthropods, bioindicators, soil quality, mesofauna, soil degradation

## Abstract

The dramatic increase in soil degradation in the last few decades has led to the need to identify methods to define not only soil quality but also, in a holistic approach, soil health. In the past twenty years, indices based on living communities have been proposed alongside the already proven physical-chemical methods. Among them, some soil invertebrates have been included in monitoring programs as bioindicators of soil quality. Being an important portion of soil fauna, soil arthropods are involved in many soil processes such as organic matter decomposition and translocation, nutrient cycling, microflora activity regulation and bioturbation. Many studies have reported the use of soil arthropods to define soil quality; among taxa, some have been explored more in depth, typically Acari and Collembola, while generally less abundant groups, such as Palpigradi or Embioptera, have not been investigated much. This paper aims to evaluate and compare the use of different soil microarthropod taxa in soil degradation/quality studies to highlight which groups are the most reported for soil monitoring and which are the most sensitive to soil degradation. We have decided not to include the two most present and abundant taxa, Acari and Collembola, in this paper in consideration of the vast amount of existing literature and focus the discussion on the other microarthropod groups. We reported some studies for each taxon highlighting the use of the group as soil quality indicator. A brief section reporting some indices based on soil microarthropods is proposed at the end of this specific discussion. This paper can be considered as a reference point in the use of soil arthropods to estimate soil quality and health.

## 1. Introduction

The capacity to define soil quality and soil health has become extremely important in recent years due to the dramatic increase in soil degradation. This topic has stimulated many soil scientists to find new methods to define soil condition and assess soil quality [1,2,3], so much so that, in the past twenty years, biological indices based on soil fauna have been developed to supplement the already tested physical-chemical methods. Alteration in soil chemical, physical, or biological properties can affect soil fauna, in terms of biodiversity, abundance and functional relationships among taxa. Accordingly, a well-developed and diversified soil fauna community is generally found in soil boasting good quality in terms of organic matter content, absence of pollution and disturbance, such as mechanical soil tillage. The main properties currently generally evaluated to characterize soil invertebrate communities are species diversity and abundance, but often they do not explain some effects of soil degradation exhaustively. Synthetic indices, such as Simpson, Pielou and Shannon indices, can integrate some information about soil fauna [4], but do not take into account the ecological role of each taxon [5], and sometimes can fail to highlight the differences in microarthropod density [6] or the state of the community structure [7].

Soil arthropods represent one of the most important components of soil-living communities and play an important role in maintaining soil quality and health and providing ecosystem services. It is a well-known fact that soil arthropods are involved in many processes such as organic matter translocation, breaking and decomposition, nutrient cycling, soil structure formation, and consequently water regulation. In addition, some groups are highly sensitive to changes in soil quality because they live, feed and reproduce in the soil, and are extremely adapted to specific soil conditions [8,9]. Among soil microarthropods, Collembola and Acari are the two most important groups in terms of abundance and diversity [10], and are also the most investigated taxa. Both groups are often investigated at the family, genus or species level, and a non-taxonomic different approach considers functional groups or functional traits [11]. Others’ microarthropod taxa that are often used to define soil quality are: (i) insects such as Coleoptera adults and larvae, Hymenoptera (ants especially), Diptera larvae, (ii) Araneae and (iii) Isopoda. However, other groups, such as Protura, Diplura, Pseudoscorpionida, Symphyla and Pauropoda, have been investigated only a little, and, when considered, they are generally discussed with a genetic, taxonomic, or ecological approach [12] rather than as soil health indicators.

This paper aims to evaluate and compare the use of different soil microarthropod taxa in soil degradation/quality studies to highlight which groups are the most reported on in soil monitoring studies and which ones are the most sensitive to soil degradation. We have reported some studies for each taxon highlighting the use of the group as a soil quality indicator. The paper does not claim to be exhaustive but rather collects and updates information about the use of this huge phylum in soil monitoring. We have decided not to include the two most present and abundant taxa, Acari and Collembola, in this paper, due to the vast amount of existing literature [10,13], but to focus the discussion on the other microarthropod groups. A brief section reporting some indices based on soil microarthropods is proposed at the end of this specific discussion.

## 2. Section for Specific Taxa

Below, we discuss the use of soil microarthropods in soil quality and health monitoring, at the order level for Hexapoda and Isopoda, and class level for Myriapoda. We have decided to maintain this different taxonomic level because these are the most used in published papers.

### 2.1. Hexapoda Entognatha

This grouping includes two little evolved orders, Protura and Diplura, in addition to the important large Collembolan group, not discussed in this paper.

#### Protura and Diplura

Protura and Diplura are groups that generally occur rarely and related to which scarce information is available. Nevertheless, their morphologic traits, such as their small chitin-less body, and their trophic functions make them potential indicators of soil stability [14]. Information regarding the ecology of Protura and their use as soil indicator is limited. Despite the many taxonomic papers published on this group [7], Protura still remain one of the less known hexapods, especially from a systematic, ecological and biogeographical viewpoint [12]. The density of Protura in soils can range from a few hundred to thousands of individuals per square meter, and their distribution is usually aggregated in relation to environmental characteristics (vegetation cover, soil pH and texture [15], and pheromones [12]. Their feeding habits are probably related to mycorrhizae, from which they take nutrients through their styliform buccal pieces [16]. There are very few studies on the use of this group as an indicator of soil health and, when considered, it is discussed in a wider approach aimed at studying the whole soil microarthropod community [17,18,19]. Moreover, Socarrás [14] observed that they inhabit deep strata, so that they are not affected by alterations occurring in higher ones. In a study that takes into consideration macro- and mesofauna in soil contaminated by oil extraction in Mexico, the authors in [20] reported that Protura were only found in control sites. The authors in [15] pointed out that some species of Protura, such as *Acerentulus cunhai* (Condé, 1950) and *Gracilentulus gracilis* (Berlese, 1908), are more tolerant to anthropogenic and degraded areas than others (e.g., *Acerentomon nemorale* Womersley 1927; *Acerentomon brevisetosum* Condé, 1945; *Acerella remyi* Condé, 1944; *Eosentomon silesiacum* Szeptycki, 1985; *Eosentomon stompi* Szeptycki and Weiner, 1993). In a study conducted in Vienna, Christian and Szeptycki [21] showed that the absence of Protura in all garden and arable sites investigated could be induced by agrochemicals or repeated mechanical disturbance. The authors stressed the effect of deforestation on the reduction in the abundance of proturan species towards the city center, assuming that an anthropogenic habitat bears poor proturan fauna.

Diplura are generally found under trucks or stones and in litter, sometimes in caves, since they avoid soil disturbance and move rapidly to better conditions [14]. In their study on different land uses in the Colombian Amazon, Suárez et al. [22] found that the Diplura group is strongly associated with native forest, reflecting the dependence of this group on places with low levels of disturbance. Moreover, this group’s requirements are found in depth, under constant conditions of water and humidity [16,23]. On the other hand, in a study aimed to define the effects of different swine manure applications on soil arthropod abundance and diversity, Diplura showed not to be affected by the type of treatment [24].

### 2.2. Exapoda Insecta

This huge class comprises many important orders involved in numerous ecological functions. Below, we discuss the orders having relations with soil.

#### 2.2.1. Coleoptera

Belonging to the biggest insects order, Coleoptera have been widely used as indicators for their rapid response to habitat fragmentation, grazing, fertilization and forest cutting [25]. They depend on several abiotic and biotic factors, and the authors in [26] divided them into generalist and ubiquitous species, species occupying a wide range of habitats (eurytopic), and specialists of one or few habitats (stenotopic). Grassland and cereal fields are the most intensively investigated among the habitats studied, as ground beetles are important predators of agricultural pests and like spiders are affected by management practices that favor some species, such as the ones that have high dispersal power (e.g., *Loricera pilicornis* Fabricius, 1775; *Nebria brevicollis* Fabricius, 1792), but not others (such as *Carabus problematicus* Herbst, 1786) [27]. In addition, since the assemblages of beetle species are strongly related to habitat characteristics, they are useful indicators for forest conservation strategies too, providing information on the organization that should be maintained in managed forests [28,29]. Beetles can be used to indicate many kinds of alteration in the environment, like pollution and post-fire recovery [30,31]. However, the ecological requirements of species vary due to their great taxonomic diversity. For this reason, at least indicator families are generally required. Some Coleoptera families show close associations with specific ecosystems, such as Cerambycidae in woodland, Chrysomelidae in foliage rich environments and Carabidae in open habitats [32].

Cerambycidae. Diversity of cerambycid beetles is affected by many factors, such as composition of tree species, canopy cover, litter and decayed trees [33]. Consequently, a change in land use, such as logging and timber extraction in the forest can affect their distribution and abundance [34]. Fahri et al. [33] noted that a polyculture area with fairly low intensity of disturbance and availability of branches, decayed wood and dense canopy cover supports high cerambycid diversity; and this diversity may be related to availability of twigs and bark as a major habitat of beetle larvae. This result supported [35], according to whom larvae of longhorn beetles play an important role in wood decay, by eating dying plants or dead wood. This food behaviour renders Cerambycid beetles biological indicators of forest areas, since they depend on availability of dead wood and are sensitive to forest conditions [36].

Chrysomelidae. Chrysomelidae species (both larvae and adults) are herbivores, considered of great economic importance due to their potential as control agents of invasive plants [37]. They are probably among the most selective phytophagous insects [38], and because of this feeding behaviour they depend on a given set of plants to survive and persist in natural habitats. This feature makes Chrysomelidae a good bioindicator, as they are expected to respond to environmental alterations such as anthropogenic disturbance. The proportion of Chrysomelidae to total Coleoptera present in an area is considered an indicator of environmental disturbance, under the assumption that herbivorous insects are less disadvantaged in disturbed habitats due to the increase in density of herbaceous and shrub plants, while predatory and detritivores insects are expected to increase in natural areas [39]. In the case of leaf beetles, Linzmeier et al. [40] found that the proportional abundance of Chrysomelidae/Coleoptera in a temperate wet forest decreases with increasing degree of succession. The authors in [41], instead, observed that this type of indicator, i.e., using either abundance or species richness, was not useful to detect major changes in the habitat structure in a low thorn forest. They noted that species richness and abundance increase in modified areas because of high plant heterogeneity due to disturbance. However, the similarity of faunistic composition between such areas and conserved sites is very low, with the conserved areas having the higher diversity. Thus, Chrysomelidae distribution in low thorn forest vegetation is unique to conserved habitats when compared with disturbed areas. Moreover, a system that includes not only species richness and abundance but also assemblage composition is needed to allow better understanding of Chrysomelidae response to environmental disturbance and, in some cases, the use of genera instead of species was found to be a more efficient tool of bioindication [41,42].

Carabidae. Changes in carabid community dominance, diversity, abundance, sex ratio, etc. have been used as bioindicators in numerous studies [43,44]. Carabid beetles constitute one of the most suitable groups for the study of ’ecological’ effects of different stressors on soil communities, as they are strongly sensitive to human induced environmental changes. Their assemblages reflect disturbance in grasslands, agricultural ecosystems and forests, with changes in composition and species abundance. Niemelä et al. [45] noted that open-habitat species abundance (belonging to *Amara*, *Bembidion* and *Harpalus*) increases in disturbed areas, together with the decrease or disappearance of forest-dwelling species. Simultaneously, disturbance causes decrease in large-sized and poorly dispersing specialist species, so that brachypterous species are more frequent in constant environment than in disturbed habitat [25,46]. Many carabid species not being strictly related to soil (usually except for the larvae cycle), some of them would be more suitable for the bioindication of surrounding environment rather than to define soil quality. However, there are some disadvantages in carabids use as indicators, for example their seasonal variation, patchy distribution and high number of generalist species [25]. Moreover, they are relatively poor heavy metal accumulators, being both holometabolic insects and predators [43].

Scarabaeidae. Due to their high sensitivity, Scarabaeidae can be good indicators of the environmental consequences of human activities and habitat disturbance too. They play an important role in maintaining ecosystem nutrient cycling and enhancing plant growth and secondary seed dispersal. Moreover, they increase primary productivity further and suppress parasites of livestock in agricultural systems. However, in temperate and tropical systems, local and regional-scale changes in land-use and mammalian communities can alter dung beetle species diversity and abundance [47]. An important result with Scarabaeidae indicators was obtained by [48], who demonstrated that re-forested habitats provide low conservation value for dung beetles and are far from being a ’conservation friendly’ solution to the increasing forest loss.

The reaction of other taxa to the change in Coleoptera communities is still poorly studied. Fattorini et al. [49] noted that the diversity of Nitidulidae, Tenebrionidae and Chrysomelidae was correlated to that of scorpions, centipedes and moths in Turkey, and other studies emphasised some resemblance in carabid and spider responses to habitat types [27,50].

#### 2.2.2. Hymenoptera

Hymenoptera have high abundance and biomass in most systems, even compared to other dominant taxa [51], and have been found to be useful ecological indicators, mainly because of their assemblage composition and richness [32,52]. Moreover, recent studies suggest that this group can be used as indicator of other arthropod taxa changes [53]. However, since Hymenoptera are represented by organisms belonging to a variety of functional groups and ecological roles, caution is needed in using higher groups (such as families) as bioindicators. Henson et al. [54] suggested that the presence of higher trophic level species such as parasitoids can be an indicator of ecosystem health, since they play a key role in food webs. Hymenopteran parasitoids are one of the most diverse group of arthropods [55], which plays an important functional role in agricultural ecosystems [56] and can be used as indicators in grassland, since abundance, family and genera richness have been found to have positive significant relationships with overall arthropod taxon richness [57]. This tool seems to be a simple method for monitoring. However, identification of parasitoids, even at high levels, requires expertise and is time-consuming [57].

At a family level, Formicidae are the most widely used hymenopteran bioindicator. Although richness of collembolan, mites, spiders and beetles species exceeds that of ants and is more used as environmental indicator, counting and identifying them is also more time-consuming than ants, which are, moreover, better indicators of assemblage composition than other groups and their ecology is also well-known [18,58,59]. Indeed, ants are social insects with stationary nesting behavior, and this aspect of their ecology allows associating them with the area where they are collected [60]. Ants have been successfully used as bioindicators in Australia [61], where their richness is known to be correlated with microbial activity in rehabilitated mine sites [62], and as indicators of pollution, forest health and rangeland condition. Previous research [63] indicated that ants respond to land changes in predicable ways, since species assemblage, abundance and richness are related to management factors, soil variables and cropping practices, so they may have potential as a biological indicator of soil condition and management in agroecosystems. Furthermore, ants play an important role as ecosystem engineers, due to their ability to increase soil drainage, aeration and nutrient quantity, thereby contributing to agricultural practices of low ecological impact [64,65,66].

Ants have a wide geographic distribution and are sensitive to several factors, such as physical and chemical characteristics of the soil [67]. For example, the number of ant species in dry soils is small, while species richness is affected by pronounced variation in humidity, and is smaller during the dry season [68]. Moreover, vegetation has also an important role in these organisms too, since reduced vegetation cover can affect different myrmecofauna groups, with negative effects on specialist species in closed and cooler areas, and positive effects on specialized species in more open and warmer areas, while increased complexity of the vegetation contributes to greater diversity and group density [69,70].

The way and intensity in which soil is used and the different succession stages can change the richness and composition of the ant fauna too. Ant communities in habitats with disturbed soils, with humidity or mineral concentration modified by anthropogenic activities, easily differ from the ones in undisturbed habitats, to which more specialized groups such as cryptic and predators are restricted [58,67,71,72]. Blinova and Dobrydina [73] found that anthropogenic impact on a given area can be measured on the basis of species richness, taxonomic structure, linear dimensions and ratio of nests types; indeed, the proportion of underground nests is used as bioindicator, since the fact of ants going into the soil is one of the adaptations to urban conditions. In terms of successional stages, habitats at the initial stage of succession harbour a myrmecofauna with a low number of species and dominance of only a few of them capable of tolerating extreme conditions; throughout the succession process, other species colonize the plots, thereby increasing diversity, while richness response depends on their being epigaeic or hypogaeic ants, since epigaeic species richness does not change with succession progression, unlike hypogaeic ants, whose richness can be higher in the intermediate and late stages of succession [74]. Stephens and Wagner [60] found that traditional biodiversity measures, such as species richness, diversity and dominance are a less effective measure of bioindication than functional group analysis, which allows considering the ecosystem role of each species. Indeed, they found that different functional groups were dominant under different levels of disturbance (e.g., relative abundance of the opportunist, such as *Formica*, *Myrmica* and *Tampinoma*, decreases with disturbance) and suppressed or excluded other functional groups that were less tolerant to these conditions. However, species composition of communities varies greatly [75] and the lack of knowledge of their taxonomy, on the one hand, and of the study of the relationship between ant species and soil physico-chemical parameters on the other can be a disadvantage in the use of ants as bioindicators. Moreover, there may be limitations owing to their resilience, since ants (but termites too) tend to easily recover once human disturbance has ceased [51,76].

#### 2.2.3. Diptera

Diptera are widely distributed, are various sizes and shapes, and represent all trophic groups [77]. According to [78], based on their relation to soil, they can be divided into three groups: (i) dwelling in soil their entire life (e.g., wingless Sciaridae and Cecidomyiidae living in soil pores); (ii) spending immature developmental stage in soil; and (iii) only pupating in soil. In temperate zones, soil-dwelling Diptera can reach 50–150 species, being higher in forests than in arable land and depending on the family they belong to, like Chironomidae in wet grassland and Empididae in formations of deciduous trees [78]. Adult stages can be highly mobile, while slowly moving larvae dwell in conditions like those of soil dwelling animals, thus the latter ones represent a more useful bioindicator of soil conditions. Larval stages play a key role in ecosystem functioning by taking part in the breakdown of dead organic matter and in nutrient cycling, and their population is highly dynamic and sensitive to environment changes [79]. Hövemeyer [80] reported that larvae abundance shows seasonal dynamics in temperate habitats and it is characterised by different vertical distribution, i.e., there is an increase in autumn and winter, and a decrease in late spring and summer, and the number of larvae generally decreases with soil depth, starting from the surface of the litter layer due to dryer conditions. Hövemeyer [80] suggested a classification of dipteran larvae based on feeding type (phytosaprophages, surface scrapers, microphages, mycophages and predators), and noted that vertical distribution can vary according to membership type, with surface scrapers abundant in the litter layer and predators frequently common in deeper soil layers. Due to their role in ecosystem functioning and their sensitivity to dry conditions (larvae are particularly sensitive to desiccation), the main natural factors affecting Diptera in soil are input of dead organic matter and soil moisture, with which they have a positive relationship [81]. However, the relationship between abundance and these factors can depend on the larval stage, with larvae searching for dryer conditions before pupation and high-water content being detrimental for the development of older larvae and pupae [82,83]. Human activities that affect soil-dwelling Diptera the most are agricultural practices, some insecticides, drainage and succession changes. Tillage is the most impacting disturbance factor in agroecosystems. Indeed, its effects are more significant than those of pesticides, affecting larvae both directly and indirectly. Removal of organic residues from the soil surface by ploughing causes direct damage to larvae and alters habitat structure (porosity and moisture content) while taking away an important food source from dipteran population and affecting predator pressure [84,85]. Other agricultural practices that can affect dipteran abundance, by increasing it, are the ones related to an overall increase in plant biomass and dead organic matter, such as mineral fertilizers and addition of manure, which enhance abundance of Chironomidae, Sciaridae, Scatopsidae and Psychodidae larvae (but also attract dipteran females and increase oviposition) [86,87,88]. The use of insecticides and/or herbicides leads to different responses in Diptera larvae, sometimes reducing their abundance and in other cases displaying no significant effect [89,90,91]. An important consequence of the application of herbicides on dipteran communities is pointed out by Eitminavichiute et al. [92]. The authors discovered that larvae could accumulate benzofosfate even when no trace of the pesticide is to be found in soil. The effects of drainage are strongly linked to moisture changes, decreasing dipteran abundance at family level, or increasing it by transforming soil from flooded to wet, but also increasing the proportion of predators and phytophages [93,94]. Succession effects depend on the dwelling Diptera, with the history of the plot before abandonment and the stage of succession benefiting some species over others [78]. Finally, phenomena that reduce competition and predation, such as fire and heavy metals, can alter dipteran community, as well as soil acidification resulting from pollution [78]. In some cases, adult Diptera are strongly related to soil condition and therefore they prove to be good bioindicators. This is the case of the Sphaeroceridae (e.g., *Spelobia parapusio* Dahl, 1909) and Lonchopteridae families, where the adults rarely fly (some species are even brachypterous) and mainly crawl around on the soil, under the vegetation, thus display reduced mobility (high site fidelity) of the species; this makes them a promising indicator for soil health [95]. Diptera can be useful indicators of soil health and management; however, due to their great heterogeneity and ecological needs, together with the difficulties of the taxonomic identification at species level, they are under-represented as bioindicators.

#### 2.2.4. Embioptera

This group is gregarious, living in tunnels and chambers woven from the silk they produce, which they rarely leave to search food. This group is poorly studied in soil monitoring research, and generally analyzed with other taxa of soil arthropods [5,96]. Domínguez-Haydar et al. [3] highlighted the importance of this group in the evaluation of reclamation success, showing that Embioptera was present in soils only after 16 and 20 years’ rehabilitation and in forests. Santorufo et al. [5] instead evaluated soil quality in urban environment and found this group in the soil with the highest total Pb and Zn concentrations.

#### 2.2.5. Orthoptera

Orthoptera distribution depends on different environmental factors, such as vegetation and microclimate, and related food availability and risk of predation [97,98]. They are considered effective indicators for many regions and habitats, particularly grassland due to (i) their sensitivity to environmental changes and land management, such as fire frequency, grazing and abandonment [99], and (ii) their role as key organisms since they are mostly under ‘bottom-up’ control, reflecting land disturbance especially when it involves the grass-layer [100]. However, their value as bioindicators appears more related to species composition than richness, the connection with which appears inconsistent among studies. In some cases, overall species richness in steppe grasslands appeared to be higher in mown and young abandoned meadows than in forests or, at least, highest in intermediated stages such as grassy heath [101,102]. On the other hand, Fartmann et al. [98] found no differences in species richness, but visible effects on habitat specialists, while Andersen et al. [100] observed a strong correspondence between species composition and degree of habitat disturbance. Nevertheless, all Orthoptera are sensitive to meso- and micro- climatic conditions [103], and their diversity and abundance correlate with vegetation structure rather than plant composition [104]. Indicator species can be used as a shortcut to identify degrading communities, and the authors in [105] found that some species of orthopteran insects are susceptible to industrial pollution and can be considered an important bioindicator group. Some species in particular are completely absent in industrial areas and may be designated as bioindicator species for industrial pollution. These are all the three species of the Tettigoniidae family (*Euconocephalus incertus* Walker, 1869; *Holochlora indica* Kirby, 1906; and *Letana inflata* Brunner von Wattenwyl, 1878), one of the two species of the Gryllidae family (*Modicogryllus confirmatus* Walker, 1859), 2 of the 3 species of the Tetrigidae family (*Euparatettix tenuis* Hancock, 1912 and *Thoracodonta* sp.), and 3 of the 7 species of the Acrididae family. These species may be used as bioindicators of industrial pollution, but some canopy species, such as the ones belonging to Tettigoniidae, can be difficult to sample.

#### 2.2.6. Isoptera

Isoptera are mainly decomposer organisms that act as ecosystem engineers improving soil structure and nutrient cycling [64]. Their value as ecological indicator is still to be proven, but they possess the attributes to be good bioindicators: widespread geographic distribution, high abundance, taxonomic and ecological diversity, low locomotor capacity, functional importance, short time response to disturbance, sensitivity to environmental conditions, easy sampling and identification [106]. Moreover, their richness can be correlated with the diversity of other groups, being useful indicators of the overall species number [107]. Some authors observed that termite species richness, relative abundance and composition are affected by habitat disturbance and fragmentation, but also by land use [108,109,110]. Pribadi et al. [111] observed that different types of land use, with various disturbance levels, affected termite species richness, relative abundance and biomass negatively, with their decreasing from protected forests to settlement areas. They pointed out three mechanisms behind the decrease in diversity observed in disturbed habitats: the decrease in humidity and increase in environmental temperature; reduced food supply and ability to nest in disturbed habitats, and, finally, increased bulk density lowering termite activity, mainly subterranean ones. They found no response of the community structure to land use, although according to Jones et al. [109], monoculture cropping system is considered a cause of decrease in the diversity of microhabitats, thus of Isoptera. Many authors find soil-eating termites to be the most sensitive group to habitat disturbance in humid forests, in that they require more stability in moist soil conditions than wood-eating termites (e.g., Rhinotermitidae, such as *Coptotermes sjoestedti* Holmgren, 1911; and *Schedorhinotermes* putorius Sjöstedt, 1896 were better represented in the disturbed sites than in the undisturbed sites) and show a gradually decreasing relative abundance in response to changes in the level of habitat disturbance, eventually disappearing [108,109,110]. However, Carrijo et al. [112] noted that pasture implantation considerably reduces abundance and richness of intermediate and xylophagous groups, since lack of trees and bushes also means less wood as food; in this context, grass/litter feeders (the ones generalist and tolerant to disturbance) can proliferate (e.g., *Procornitermes araujoi* Emerson, 1952, *Syntermes nanus* Holmgren, 1909 and *Ruptitermes* sp.). Nevertheless, a direct response can be observed between local termite diversity and conservation of the area, so, depending on the study area, absence of a termite group compared to another could be used as bioindicator of environmental quality [111,113].

### 2.3. Arachnida

This is a numerous and diversified class that comprises some important orders present in soil or litter, such as Pseudoscorpionida, Araneae (spiders), Opiliones (harvestmen), Acari and Palpigradi. In this paper, we discuss Pseudoscorpionida, Araneae, and Opiliones only. The microwhip scorpion (Palpigradi) is not discussed because we only found a morphological-anatomical description and information about its feeding habit [114], but no literature on the use of this group as bioindicator of soil quality. Acari instead are not treated because of the extensive research published on this group.

#### 2.3.1. Pseudoscorpionida

This group is generally present in soil grassland and wood, where cover vegetation or litter is present. They are carnivorous arthropods and important predators in generally stable environments. Few studies report the use of this group as soil quality indicator, and often it is discussed together with other members of arthropod community [2,20]. In a study aimed to evaluate the effects of different swine manure applications on soil arthropod abundance and diversity, Pseudoscorpionida were more abundant in the injection treatment throughout most of the post-manure application period [24]. Çakir [6] showed that a natural disturbance, such as the one caused by the ant *Formica rufa* (Linnaeus, 1758), affected the abundance of this group negatively. Çakir [6] hypothesized that this effect might be brought about not only by the predation of *F. rufa*, but also by competition between predators. In the study reported by Domínguez-Haydar et al. [3], Pseudoscorpionida, such as Embioptera, were present in the more aged restored areas.

#### 2.3.2. Araneae

Spiders have occasionally been used as bioindicators, and in recent years some studies have underlined their potential as good bioindicators [115,116]. First, they are characterised by great taxonomic diversity and abundance in both natural and cultivated environments, with average annual abundance ranging from 50 to 150 individuals per square meter [117]. Moreover, they respond to environmental changes, and ground-dwelling species are highly sensitive to even small changes in habitat structure (mainly related to vegetation) and microclimatic factors, which may be altered by anthropogenic disturbance [115,118,119,120]. Both vegetation and leaf-litter structure (depth and composition of plant debris) determine specific microclimatic conditions whose variations determine changes in spiders’ specific dominance, since they are heterothermic arthropods [121]. Burned sites tend to be characterized by open habitat, non-web-building species (e.g., *Gnaphosa borea* Kulczynski, 1908; *Pirata bryantae* Kurata, 1944; and *Arctosa alpigena* Doleschall, 1852), often belonging to active hunting families that are gradually replaced by web-building families as vegetation regenerates [122]. Pearce and Venier [120] noted that spider assemblage provides evidence that anthropogenic disturbance (harvesting and silvicultural activities) differs from natural disturbance (wildfire) and can therefore be useful to determine the sustainability of forest management practice. Araneae give information not only about vegetation structure and microclimate variation, but also about arthropod community, being among the top macroinvertebrate predators. Moreover, the amount of prey ingested by spiders depends on the total quantity of potential prey, thus can give an indication of the biological quality of the habitat, as changes reflect on spider composition and dominance. For example, structural complexity created by vegetation and litter cover (like during afforestation processes) can provide diversity of prey, thereby increasing spider species richness [119]. Ysnel [123] suggested that the diversity of prey ingested in the field might also be correlated to the sexual proportion, as it was noted that under the same temperature conditions, more females reach adulthood when fed with a variety of prey. Spider population can also be affected by human disturbance through exposure to high concentration of heavy metals and habitat fragmentation [124]. As consumers, spiders are involved in biological magnification of a variety of contaminants. Some studies have confirmed that heavy metal concentration in spiders can lead to a field evaluation of ecosystem contamination level, above all ground-dwelling spiders, which feed on relatively immobile prey items within the contaminated field plots and live in direct contact with contaminated soil [125,126]. Another stress factor reflected by spider populations is habitat fragmentation, resulting from intensification of agriculture and urbanization that may cause local extinction of small populations. Web-builders and wandering spiders generally aggregate in areas with abundant supply of prey; however, this aggregation can be strongly limited by competition among spiders, or by parasitism and predation upon spider populations [127]. Pearse [117] summarised the main advantages of using spiders as bioindicators as follows: (i) they are present at high densities, (ii) they exhibit specific ecological demands toward their natural habitat, (iii) community variations can be detected even for a small area within a given biotope, and (iiii) they hold a strategic level within the food chain as predators or prey.

Several methods have been formulated to analyse spider community structure in order to evaluate human disturbance or natural habitat quality, but they appear to be most effective in indicating changes on small spatial scale [120]. These studies usually use a group of species or families as indicators and, only occasionally, a single species shows a connection with a specific ecological condition [115]. Generally, the reasons why a spider species is present or excluded from a given habitat are still poorly understood, and identification is difficult within some families. Moreover, when considering taxa as bioindicator, the dispersing abilities and great ability to resist adverse conditions of Araneae should be taken into consideration.

#### 2.3.3. Opiliones

Opiliones have worldwide distribution and represent the third largest order of arachnids [128]. They are carnivorous arthropods and important predators in agriculture, where low-input crop fields yield many parasitic insects and represent a great resource pool, although great species richness is found in forested landscapes [129,130]. They are considered invertebrate bioindicators with high niche specialization and ecological requirements, therefore vulnerable to environmental disturbance [131,132]. Their spatial distribution appears to be highly dependent on environmental complexity, with some species showing microhabitat plasticity and others’ microhabitat specificity [131]. Mihál et al. [133] found that Opiliones can in turn be divided into three groups according to their ecotrophic and ecotopic specialisation (forest stand, ecotone stand and open habitats), since they easily respond to abrupt changes to the forest environment. The predominant local richness in most temperate habitats ranges between four and six species [130], with *Phalangium opilio* (Linnaeus, 1971) frequently found in agricultural habitats, being a polyphagous and pests feeder species. Drummond et al. [134] observed that, in lowbush blueberry, *P. opilio* appeared to be less dominant in organic than conventionally managed fields, corresponding to an increase in the number of *Rilaena triangularis* (Herbst, 1799). In general, the dominant harvestmen are synanthropic species and Curtis and Machado [130] concluded that they survive better in disturbed habitats, while specialist and rare species show high vulnerability [110]. However, even if disturbing agents like fire can have detrimental effects on these communities, their recovery is rapid [135]. In addition, their use as ecological indicator is hindered by difficult identification and taxonomy.

### 2.4. Myriapoda

This subphylum comprises four classes, Chilopoda, Diplopoda, Pauropoda and Symphyla. The two former classes are sometimes used as bioindicators in soil monitoring projects. Few studies have reported the use of Pauropoda and Symphyla as soil quality indicators, generally not alone but together with other taxa of soil arthropods.

#### 2.4.1. Chilopoda

Chilopoda are carnivorous arthropods present in different habitats. Despite being very active and mobile, this group has been used in habitat quality indication [136]. Chilopoda are mostly generalist predators, especially abundant in soil and litter of forest and agroforestry systems [137], where they play important ecosystem functions, such as regulation of decomposer population [138]. Changes in centipede communities depend on changes in habitat and therefore in prey abundance. Thus, a reduction in litter layer can affect Chilopoda populations, but also pH. Klarnel et al. [137] observed that Chilopoda communities are affected by increasing pH in the litter layer as a result of the pH sensitivity of potential prey taxa, such as earthworm and enchytraeid species. Moreover, they found that, due to their ability to switch to alternative prey, Chilopoda are unable to acclimate to alteration in the environment, despite showing changes in community composition.

Community structure and diversity of centipedes are affected by many variables, the most important of which are temperature and humidity [139]. In some cases, however, like in hedgerows in agricultural land, species richness can also be influenced by the nature of surrounding habitats (e.g., sites surrounded by arable land support poor species assemblage) [140]. Even soil structure influences Chilopoda communities, especially the abundance of groups that spend most of their life cycle under the soil and may depend not only on the degradation rate of organic matter, but also on the possibility of digging for shelter to avoid dehydration, like Geophilomorpha and Diplopoda [141]. Moreover, myriapods are good indicators of biological soil quality, as they need time for immigration and can be used to assess success in site rehabilitation. Dunger and Voigtländer [142] studied Myriapoda colonization of mine sites over a very long period and noted that the first establishment is influenced by surrounding fauna, invasion capacity, dispersal ability and life strategies. They observed that Chilopoda pioneers (e.g., *Lamyctes emarginatus* Newport, 1844) depend on the very early immigrating microarthropods that serve as prey, but generally they are also characterized by other factors that enable them to tolerate stress, like short generation cycle, parthenogenetic propagation and high mobility. Geophilomorpha instead are predators hunting within the soil pores, characteristic of ‘woodland-like stages’, and give information about the A-horizon (e.g., *Strigamia crassipes* C.L. Koch, 1835; *Strigamia acuminata* Leach, 1815; and *Geophilus alpinus* Meinert, 1870). The most important result in this study is that many species of natural centipede assemblages (mainly belonging to the genus *Lithobius*, such as *Lithobius dentatus* C.L. Koch, 1844; *Lithobius nodulipes* Latzel, 1880; and *Lithobius macilentus* L. Koch, 1862), typical of the region studied, were still lacking even in rehabilitated sites (which look natural).

#### 2.4.2. Diplopoda

Diplopoda is a common group generally present in wood but also in grassland and in agriculture ecosystems. This group is affected by soil moisture reduction since they are subject to fast desiccation. They move in depth to lay eggs or moult in the dry or cold period.

In a study conducted in urban soils in Italy, Santorufo et al. [5] suggested that Diplopoda could be considered a ubiquitous taxon, since it was observed in all the soils studied except in the soil with the lowest metal contamination. Huot et al. [13] reported relatively high abundance of this group in an industrial setting pond managed by natural attenuation. The authors reported 39.9% relative abundance of Diplopoda within all macroinvertebrate collected, and within this percentage 37.8% were Julidae (*Cylindroiulus punctatus* Leach, 1815; *Julus scandinavius* Latzel, 1884; and *Ommatoiulus sabulosus* Linnaeus, 1758). Diplopoda are detritivores, therefore their assemblage composition, species diversity and population density correlate with vegetation structure [143]. For this reason, they are associated with more stable environments, with a significant relationship between soil nutrient quality and microbial activity, helping to mineralize nutrients and making them available to plants [144]. Like Chilopoda, millipedes are affected mostly by temperature and humidity [139]. Bogyó et al. [145] found that millipede abundance follows the edge effect hypothesis, being highest in the forest edge, perhaps because they find higher temperatures and openness there, as well as leaf litter cover, dead wood and soil moisture. However, the edge effect is not evident in species richness and diversity, and assemblages change with vegetation structure, being the habitat type even more important than successional stage [146]. Macroporosity was important for saprophytic groups too, and was related to soil microbial activity and abundance, and to soil and litter chemical quality, affecting the interaction between Diplopoda and soil microbial organisms [147]. As for Chilopoda, Dunger and Voigtländer [142] studied immigration of Diplopoda in post-mining sites too. True pioneer millipede species do not exist; therefore, they distinguish between “early” (e.g., *Craspedosoma rawlinsii* Leach, 1814; and *Praemurica inconstans* Subbotina, 1953) and “late” colonisers (in this case, mostly species preferring woody habitats, e.g., *Polyzonium germanicum* Brandt, 1837; and *Melogona voigtii* Verhoeff, 1899). Even in this case, characteristic species (especially *Strongylosoma stigmatosum* Eichwald, 1830), frequently occurring in the surroundings, are still absent in the oldest mine sites, indicating that succession to typical woodland is not complete. Species numbers and abundances increased in relation to site conditions and quality of food. As for other saprophagous, unsuitable soil conditions are low water retention, low content of organic matter and high acidity [148], while pine needles are unsuitable food resources. Moreover, species composition depends on the surrounding habitats. Stašiov et al. [140] noted that Diplopoda richness in hedgerows depends on dominant tree species, as well as the nitrogen content of humus. Generally, Diplopoda seem to be good bioindicators, able to discriminate between areas with different recovery ages, even if their significance for ecological restoration is not really understood yet.

#### 2.4.3. Pauropoda and Symphyla

Pauropoda and Symphyla are two groups often reported together within the microarthropod community. Both groups generally show high sensitivity to stress in soil habitat and generally characterize undisturbed soils [149,150]. 

Palacios-Vargas [23] noted that Pauropoda are very sensitive to agricultural practices, in correspondence of which population can decrease to 70%. Bedano et al. [150] suggested that the reduction of pauropods density in high-input management systems can be largely explained by mechanical and chemical perturbations produced by conventional agricultural management practices, as well as by unfavourable abiotic soil conditions present in intensively managed sites. Moreover, in a study on a soil contaminated by oil extraction, Garcia-Segura et al. [20] reported that Pauropoda were only found in the moderately contaminated sites, while Symphyla were detected in unpolluted sites only.

Bedano et al. [150] also noted that Symphyla density was negatively affected by high bulk density, suggesting that soil compaction may influence the occurrence of these animals in agroecosystems. In addition, SOM content may be another factor affecting symphylan populations, since a positive correlation between symphylan density was found in the same study, although data on both myriapod groups were variable, so definite conclusions cannot be drawn.

### 2.5. Crustacea

Most crustacean species are aquatic. Isopoda is the order that has mostly colonized emerged lands, but its distribution is generally driven by soil moisture.

#### Isopoda

Isopoda are worldwide organisms, distributed from deserts to forests, most of them living in soils and litter layers and so being abundant in grasslands. However, a consistent difference in species composition between woodlands and cultivated areas is still noticeable [151]. Isopod species richness was found to increase with the increase in air humidity, which is pivotal for their survival given that they need it to breathe and reduce body water loss [152]. Another key factor is food availability; their feeding habits are mostly based on decaying organic materials, with preference linked to leaf senescence, nutrient content of food and microbial colonization, so that their richness tends to increase with woody plant richness and soil organic matter [151,153]. Due to their ecological behavior, terrestrial isopods play an important role in decomposing leaf litter and mineralizing organic matter in many agroecosystems. Moreover, they have been used as bioindicators of soil quality and biodiversity to improve both agricultural production and environmental protection [152,154,155]. Isopod diversity, species richness and sex ratio (e.g., in populations of *Trachelipus rathkei* Brandt, 1833) are known to be affected by agricultural practices (e.g., ploughing, fertilization and pesticides) because of their direct and indirect effects on food availability, humidity and pH [151,155,156,157]. Some studies noted that species diversity and abundance decline in intensive agricultural systems; above all, differences have been observed between organic and conventional management, where hedgerows have proved to be important in maintaining biodiversity providing a refuge and a source of woodlice for Isopoda community [151,158]. Vignozzi et al. [149] reported a higher presence of this group in the less impacted natural cover soil than in the conservational tillage soil, highlighting the high sensitivity of Isopoda to habitat stress. Different grassland management has different impact on Isopoda, which are sensitive to botanical composition not only for food but also for vegetation structure, with increased mortality resulting from simplification of the habitat [159]; so that cutting, for example, is a non-selective method that differs from grazing, where impact depends on the herbivores present [158,160]. Cultivation employed in the preceding year is also an important factor affecting Isopoda population [158]; moreover, depending on their biology, different species may be used as indicators of different factors, with some species being more sensitive to pH (e.g., *Armadillidium vulgare* Latreille, 1804; and *Oniscus asellus* Linnaeus, 1758) and others to variation in temperature (e.g., *Ligidium hypnorum* Cuvier, 1792; *Philoscia muscorum* Scopoli, 1763; and *Porcellium conspersum* C. Koch, 1841) [156,161].

Isopods are an interesting group of arthropods even because of their rapid response to contamination and their ability to accumulate high levels of heavy metals [162]. As reported for Diplopoda, Santorufo et al. [5] highlighted the presence of Isopoda in the investigated soils with higher metal content. The authors suggested that the presence of this taxon in contaminated soils might be affected by the entire set of soil characteristics and not only by metal presence. However, previous studies [163] noted a significant correlation between metal concentration in isopods (e.g., *Porcellio scaber* Latreille, 1804) and, in the substrate, so that they can be considered useful bioindicators of metal pollution, as well as an alert of the possible spread of contaminant at higher trophic levels, since they are prey of different animals (e.g., spiders, chilopods, ground beetles and others) [151].

## 3. Indices of Soil Quality Based on Soil Arthropods

Several indices based on soil fauna have been developed and proposed in the past two decades. Among them, numerous studies have used arthropods as bioindicators, generally focusing on low taxonomic level (species or genus) or on specific group, typically Collembola and Acari. The ratio between these two major groups has been used as soil indicator in some studies [5,164]. Generally, the greater abundance of Acari compared to Collembola suggests good soil quality and habitat stability, but Santorufo et al. [5] highlighted a different result in metal contaminated soils.

Some authors proposed indices based on high taxonomic level (order, family, and in some cases class), or based on more systematic taxa [10], applying several ecological indexes [165]. Cortet et al. [166] compared two crop systems which differed for tillage and pesticides application, and applied AV biodiversity index, considering microarthropod taxa separated in oribatid mites, other mites, collembolas, and other arthropods. Nuria et al. [1] proposed an integrative approach named IBQS (Indice Biotique de la Qualité du Sol) to evaluate soil quality considering assessment of the soil macro-invertebrate communities directly involved in soil processes that guarantee ecosystem services. The authors showed that the IBQS index was affected by the intensity level of management practices. Andrés and Mateos [96] reported an extensive study applying several approaches: density, different taxonomic level, percentage of some sub-groups (e.g., Mesostigmata), percentage of predators, and species composition (for Collembola and Acari Oribatid), while García-Segura et al. [20], considered a large grouping comprising macrofauna and mesofauna.

Considering soil macroinvertebrate community, and chemical and physical soil parameters, Domínguez-Haydar et al. [3] applied the General Indicator of Soil Quality (GISQ), a synthetic indicator of physical, chemical and biological quality, to evaluate reclamation success in an open-pit coal mine in a long-term study. Considering taxonomic richness and GISQ, the authors tested how the chemical and biological indicators changed with the age of the reclamation. They concluded that soil quality increased along the cronosequence, but differently in relation to the sub-indicator considered. As for soil macroinvertebrates, the study showed that the largest part of the biodiversity had recovered after six years.

Among the several indices developed in the past few years, QBS-ar (Soil Biological Quality-arthropod based on Biological Forms approach) index links biodiversity of soil microarthropod community to the degree of soil vulnerability [8]. Up to now, several papers have reported the results of the QBS-ar application, and elaborating the QBS-ar data reported in numerous papers, Menta et al. [167] highlighted the relationship between soil quality, expressed as QBS-ar, and different land uses. In a study aimed to evaluate the effects of two soil conservation management practices in a long-term (10 years) olive orchard management, Vignozzi et al. [149] reported a multiple approach. The authors determined BF (Biological Forms) richness, Shannon (H’), Margalef (d), QBS-ar index and feeding morphotype diversity classification. Recently, Çakır [6] has applied QBS-ar to assess the impact of natural disturbances caused by *F. rufa* wood ants. The author showed that this index described the competing relation between soil arthropods and ants better than other indices, such as Shannon diversity index. To explain this result, the author argued that the adaptation level of microarthropods to soil is an important element in QBS-ar computation.

Menta et al. [2] compared QBS-ar [8] and IBS-bf [168], both based on soil fauna community but different in methodology and in the taxa involved, and recorded a similar trend of the two indices in organic and conventional agricultures, highlighting the most stressed soil condition in the latter management.

## 4. Conclusions

Soil arthropod diversity, distribution and abundance are determined by several factors such as organic matter content, soil features, cover vegetation, soil disturbance (mechanical soil tillage), fire, and pollution. Moreover, the different taxa respond differently to a variety of environmental factors. Some groups are more sensitive, while others are ubiquitous and more able to react to soil degradation. In addition, different species belonging to the same taxonomic group can respond differently. Because of this complex scenario, the subject is particularly complex, as found in a wide number of papers published on this topic. As is known, Acari and Collembola are the two groups which are generally considered in soil quality evaluation approach. Other groups, such as Coleoptera, Diptera and Araneae, are often involved in studies aimed to evaluate soil quality/degradation/pollution, while other groups, typically Symphyla, Pauropoda, Pseudoscorpionida and others, are generally little discussed in soil quality monitoring or discussed together with other soil arthropods taxa. Finally, the use as bioindicator in soil monitoring programs is still missing for some groups, such as Palpigrada. In our opinion, further studies are needed to develop the ecological knowledge of the more sensitive groups and understand how the whole soil arthropod community is able to react to soil perturbation and recover.

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
