# Peer review of "Soil Health and Arthropods: From Complex System to Worthwhile Investigation"

_insects, 2020, doi:10.3390/insects11010054_

Round 1

Reviewer 1 Report

comments by line

35/36: the most important factor affecting soil fauna is not mentioned: mechanical soil tillage; instead, degradation is not a reason for reduced soil fauna, decreasing soil biodiversity is a sign of degradation. The most frequent reason for decreased soil biodiversity and hence degradation is mechanical soil tillage, while pollution and chemical imbalances are less frequent. Pls. correct

600: pls. highlight soil disturbance (mechanical soil tillage) and fire as important factors besides the ones mentioned; those two are so important, that they should not be listed as “etc.”; in fact, they are much more present worldwide as factors than, for example, pollution.

Author Response

Dear Reviewer 1,

Thank you for your positive opinion and for your comments. We have modified the MS considering your suggestions carefully.

Best regards,

Cristina Menta

Reviewer 2 Report

This study was designed to evaluate and compare the use of different soil microarthropod taxa in soil degradation/quality, and was well designed and carefully conducted. The results are addressing import questions about most sensitive groups in soil monitoring studies. This paper is interesting, and the findings of this study are worthy of publishing in this journal. However, I have few minor concerns regarding the suitability of publication for this manuscript.

Title: “microarthropods” in the title is not accurate. “Coleoptera, Hymenoptera, spiders…….” are evaluated in this study, but body length of these groups is much bigger. I suggest change “microarthropods” to “macroarthropods”. “Microarthropods” usually contain springtails and mites, which body length is less 2mm.

Line 24-25: add a sentence to discuss implications of your findings.

Author Response

Dear Reviewer 2,

Thank you for your positive opinion and for your comments. We have corrected the MS following your suggestions carefully.

Title: We agree with your comment and we have consequently decided to maintain only “arthropods” in the title (because we discussed taxa belonging to micro- and macroarthropods). We have added “microarthropods” and “macroarthropods” in the key words.

L 24-25 We have added a sentence to consider the implication of our paper as you suggested.

Best regards,

Cristina Menta

Reviewer 3 Report

Dear authors, I would like to congratulate you on the work presented.
This article represents an excellent reference point for the academic bibliography on the topic of soil arthropods applied to the soil quality test. The work is synthetic but complete.
I was able to trace only some points where it is appropriate and necessary to dwell to complete the document.

Comments can be grouped into three types:
- numbering and style of the text sub-division;
- mentioning the most important genus and species for all the taxa considered;
- insertion of some completions to the contents of the paragraphs. In attach you can find the file with the proposed comments.

Thank you for your useful work.

Author Response

Dear Reviewer 3,

Thank you for your kind comments. We evaluated every point and modified the MS considering your suggestions carefully.

We have corrected the distribution and numbering of the paragraphs in the manuscript. We realized that the original version of the MS was correct and the Conversion System had modified the numeration. We hope that the conversion process will not change the numbering again. We have changed the title of the divisions to the chapters following your suggestion.

L 121 and 153 We have added a sentence as you suggested.

L 157 We have followed your suggestion but we have preferred to insert “agricultural ecosystems” instead of “cropfield”.

Arachnida paragraph. As you suggested, we consulted some Duso’s published papers but we found only papers related to spider mites (Acari). In our MS, we did not discuss Acari. If the Reviewer would like to suggest us papers by Duso related to the other orders discussed in our MS, we will be happy to insert them in the MS.

Paragraph 2.4.1. L 530

As suggested, we have added “sex ratio” and the corresponding Reference in the text and in the References section.  

Best regards,

Cristina Menta